# Deciding on a Continuum of Equivalent Alternatives Engaging Uncertainty through Behavior Patterning

Kusal Rathnayake [1,*,†] , Alexander Lebedev [2,*] and Dimitri Volchenkov [1,*,†]

1 Department of Mathematics and Statistics, Texas Tech University, 1108 Memorial Circle, Lubbock, TX 79409, USA
2 Institute of Psychology of Russian Academy of Sciences, Yaroslavskaya str., 13, 129366 Moscow, Russia
* Correspondence: kusal.rathnayake@ttu.edu (K.R.); lebedevan@ipran.ru (A.L.); dimitri.volchenkov@ttu.edu (D.V.)
† These authors contributed equally to this work.

**Abstract:** A psychology experiment examining decision-making on a continuum of subjectively equivalent alternatives (directions) revealed that subjects follow a common pattern, giving preference to just a few directions over all others. When restricted experimental settings made the common pattern unfeasible, subjects demonstrated no common choice preferences. In the latter case, the observed distribution of choices made by a group of subjects was close to normal. We conclude that the abundance of subjectively equivalent alternatives may reduce the individual variability of choices, and vice versa. Choice overload paradoxically results in behavior patterning and eventually facilitates decision predictability, while restricting the range of available options fosters individual variability of choice, reflected in almost random behavior across the group.

**Keywords:** choice overload; behavior patterning; Gaussian mixture modeling; stochastic neighbor embedding





## 1. Introduction

*Artificial intelligence* (AI) is believed to demonstrate rapidly growing capabilities to predict human behavior, to identify vulnerabilities in human habits, and to use them to steer human decision-making through manipulative interactions [1]. Countless studies discussing the use of AI to understand how to work with humans suggest that human choice is *anything but random*, and is predetermined by personal traits and attributes hidden in behavioral complexity that can be, nevertheless, discovered by sophisticated AI algorithms through the automated identification of patterns in digital records of individual behavior.

For example, Facebook "likes" were found to accurately predict sensitive personal information, such as sexual orientation, ethnicity, religious and political views, personality traits, intelligence, happiness, addiction, parental separation, age, and gender [2]. The information obtained by tracking a user's liking activity may be used for personalizing custom-picked content to nudge a person toward particular actions. According to the Center for Humane Technology [3], the attention of an entire nation can be purchased for the price of a used car, as around 68% of adults in the US are using Facebook for posting pics, shopping, identifying places where they can dine, creating events, and sharing awareness [4].

Indeed, AI algorithms are great at solving specific problems, but only as long as you stick to the script. My own (D.V.) struggle with Facebook has been going on for many years. My courtesy requires me to like every single post of my friends. My likes are a simple sign of respect for a dear friend and do not express my actual (dis)agreement with posted texts (that I almost never read). Due to my unusual liking strategy, I was regularly misidentified as a bot , distracting the deep-learning process in the platform's recurrent neural networks, and, therefore, was blocked for liking too many friends' pages. Later, they began to set a

quota for the number of likes I could use, which I usually filled up in a short amount of time to be blocked again. Then, they tried to hide new posts in my friends' feed from me. In response, I searched for my friends' pages by name and liked all their posts in a row. After a while, Facebook gave up and started showing me some random ads, accompanied by a question as whether they met my interests and needs. Needless to say, I deliberately ignore these questions and diligently block the advertisers.

All this would be funny if we were not continuously exposed to a great many of unsolicited manipulative experiments involving a variety of AI algorithms, many of which can be seen as similar to hypnotic suggestions, for which none of us gave any consent. While some innocuous AI-based algorithms may improve our lifestyle by creating healthier dietary habits, others might massively compromise democratic procedures through election meddling; hack the national economy by steering the responsible decision makers towards accepting faulty, biased, or malicious policies; and even impel people craving for belonging to join military service in the course of unconstitutional mobilization for fighting in a neighboring country against which the war has not even been declared. Who should then be held accountable for the war crimes enthusiastically committed by soldiers indoctrinated by irresistibly convincing AI-propaganda? To what extent are we liable for our decisions and actions if manipulated by algorithms? What degree of responsibility for the crisis should be borne by the AI developers and social media platforms that implemented these "*weapons of math destruction*" [5] worldwide?

In modern psychology and especially in neuropsychology, there are ongoing discussions about the extent to which human decision-making under uncertainty may be predicted and inferred [6,7]. Drawing a decisive conclusion about possible psychological mechanisms beyond making a choice would be important for solving many scientific and practical problems, such as the philosophical problem of free will and determinism motivated by concerns about moral responsibility for our personal actions, understanding scanning and searching activities in humans and animals, predicting an individual's investment behavior, etc. Perhaps the most common type of uncertainty is choosing between two or more subjectively equivalent alternatives, illustrated by the famous parable of Buridan's ass.

One may believe that making a choice under uncertainty of many equivalent alternatives can involve some random, as well as deterministic, actions. In many studies, e.g., [8–12], it was demonstrated that, paradoxically, the *more* subjectively equivalent alternatives available for a subject, the *less* random the choice made seems to be (when observed in a group of subjects). Confronting a multitude of subjectively equivalent alternatives, humans may try to reduce uncertainty of choice by following some common choice patterns. In our work, we support this observation and show that it also appears true the other way around. Namely, by restricting the range of available subjectively equivalent alternatives, we may achieve increasing variability in decisions made by a group of subjects.

After a review of results on and discussions on whether humans can be random in Section 2, we report on the results of a triple psychological experiment, in which three gender-balanced groups of subjects were offered to make a choice on a continuum of subjectively equivalent alternatives (directions) (see Section 3 for further information). The analysis of experimental data with the use of machine learning algorithms (described in Section 4) shows that while making their choice, subjects followed a common pattern, giving a preference to just a few directions (featured by the main compass axes, as discussed in Section 6) over all others (Section 5), although the individual strategies implemented to fulfill the common pattern may greatly vary (Section 5.1). By restricting the experimental settings further (Section 3), we broke down the common pattern observed in the first experiment, making subjects follow individual patterns of choice on a continuum of equivalent alternatives (Sections 5.2 and 5.3). The experiments revealed no gender-specific differences in the random decision-making processes. We conclude in the last section (Section 6).

## 2. Can Humans Be Random?

This question is obviously still up for debate [13–15]. Research suggests that when it comes to random thinking and decision-making, a person's ability peaks at about age 25 and then gradually declines for the next several decades, eventually dropping off sharply at around age 60 [16]. Other researchers proposed that while selecting at random, people may unconsciously pattern their behavior, albeit, subjectively, they do not realize it. In the famous shopping experiment [9], subjects preferred to take the (last) *fourth to the right* item in a row while choosing between four identical products in a supermarket. In another study [12], subjects mainly crossed out cells *at the center* of ladder and pyramid figures, or distributed the assets of 11 funds *clockwise* among 12 unfamiliar managers sitting at a round table.

Little is known about the nature and mechanisms of random behavior in humans and animals although flipping a coin is proven to be an effective and time-saving decision-making strategy, especially when time to decide is rather short, decision outcomes are practically indistinguishable or ambiguous, or when under severe uncertainty [17,18]. Although deciding at random is intentionally suboptimal [18], the ability of living organisms to make random decisions rapidly under conditions of uncertainty may be important for survival [17–20]. Mimicking a random process is a mentally difficult task, requiring sustained and focused attention [16]. Humans cannot consciously generate random number sequences [21].

Nevertheless, psychologists traditionally believed that the ability to make random decisions is inherent to humans and all living organisms, which manifests in scanning environments even in lack of stimuli [22]. Until the end of the 1960s, most psychologists (K. Coombs, D. Pruit, W. Edwards, R. Lewis, and others) believed that people adequately perceive the probabilistic nature of tasks associated with risk, intuitively apply the notion of probability, and follow the conclusions of probability theory in everyday life [23]. However, in numerous psychological experiments involving mathematics majors well versed with the theory of probability, it was found that most of them did not apply probabilities in everyday life, rather making irrational decisions in pretty much the same way as those who were not familiar with this theory [18,19]. Professional investors and risk management executives also commonly disregard probabilistic wisdom about what they should not do and fail to recognize that mathematical equivalents can be psychologically different [24].

Some studies assume that the ability for making decisions at random is given to all biological organisms by evolution [11]. In an evolutionary context, randomness in searching behavior and wandering with no incentives is a necessary part of complex cognitive activity critical for survival, such as foraging and hunting [11,25]. The Lévy foraging hypothesis [20,25] suggests that natural selection should have led to adaptations for random foraging movements in animals, fitting the statistics of Lévy random flights and walks [26], as they optimize searching efficiency in the absence of memory (at least in one dimension). Namely, wandering based on an inverse square Lévy random walk consisting of a constant velocity search following a path whose length is distributed over an inverse square Lévy stable distribution is optimal for searching sparsely and randomly distributed re-visitable targets [27–29]. Indeed, saltatory searching trajectories composed of consecutive displacements $l$ drawn from a power-law distribution, $P(l) \propto l^{-\mu}$, with the scaling exponent approaching the theoretically optimal value $\mu = 2$ would maximize a forager's chance to locate sparsely and randomly distributed prey, and therefore may be an evolutionary beneficial strategy to spatially intensive search [20,29,30]. Interestingly, movements resembling Lévy flights have been identified over large groups in many living species, ranging from micro-organisms to humans, although the reported scaling exponents vary substantially for different animals and in different environmental contexts [31–43].

However, while distributions of displacements for the population aggregate do fit the power-laws, $l^{-\mu}$, an individual's bout distributions do not [44,45]. Detailed data analysis shows that movement lengths within individual tracks rather fit an exponential distribution, and therefore the power law (with an exponential cut-off) resulting from a superposition of

many exponential distributions with varying parameters in the population [45–47]. Thus, the mysterious "random intelligence" observed in large groups of animals and humans turned out to be just a game of chance.

Treasure hunting experiments with a gender-balanced group of human participants browsing unknown virtual environments in search for collectible objects [48] demonstrated that a random balancing of exploitation and exploration modes of behavior ("*should I stay*", or "*should I go*") may be responsible for the power-law statistics of displacements and duration (i.e., acts of commission and omission). As subjects participating in an experiment and acting in virtual environments amid uncertainty, any pre-cursive calculation of an optimal strategy [49] was impossible for them. The detailed analysis of quick scanning body turns (200–300 ms) being the essential part of the adaptive movement strategy under reduced natural multisensory conditions in a virtual space gave us conclusive evidence of that the total reorientation duration is strongly reinforced with the net displacement of subjects, which makes the intensive scanning process biologically unfeasible and time consuming on large spatio-temporal scales.

In the absence of any cues marking the location of hidden treasures, subjects searched in a saltatory fashion; they marched along corridors and across halls, paused for the local search in the nearest rooms, and then resumed traversing the environments. A simple, analytically tractable stochastic mechanism of the coherent noise type ([17], Ch.1) describing the random exploration–exploitation trade-off in humans, in a form of the recurring comparison of expected chances to find a treasure in an immediate neighborhood and to be rewarded in other parts of the environment yet to be explored, that fits the experimental data well was suggested in [48]. The main outcome of the exploration–exploitation dilemma as discussed in [48] agrees with the conclusion of R. Nisbet and T. Wilson that people are practically unaware of the fact that they make decisions based on things that accidentally catch their eye or thoughts that suddenly come into mind [9].

Possible mechanisms of appealing to chance while facing subjectively equivalent alternatives remain an important open problem in experimental psychology that may help to determine a baseline of cognitive function [50]. On the one hand, a random choice may occur as a manifestation of some innate ability possibly not reflected by human consciousness. In the latter case, the distribution of choice made by the group at random should be relatively *even* between all possible alternatives, regardless of their number and relative position.

Many experiments have shown that when deciding on subjectively equivalent alternatives, people often follow the Laplace principle, considering these alternatives as equiprobable [10]. For instance, subjects tend to attribute equal probability to the winning of each candidate in a tournament of four equivalent challengers [8]. On the other hand, an *apparently random* choice can be based on patterns fostered by some structural properties of the environment and experimental task settings, or by some memorable events that happened in a person's life. Being in a situation of random choice, different subjects may therefore focus their attention on the diverse features of seemingly similar alternatives; then, we could discover some common choice patterns by analyzing records made by a group of people. If the choice is made by applying some subjective strategy, it is not entirely clear whether any positional effects would persist with an increase in the number of alternatives; however, such an effect would be context dependent; for example, it might be conditional on information presenting. In order to address this question, we designed and conducted the following psychology experiment.

### 3. A Triple Experiment on Choosing on a Continuum of Equivalent Alternatives

A researcher carrying out psychological experiments often cannot control all variables for objective reasons; the experimental hypothesis may not be formulated unambiguously or experimental control groups may not be comparable at baseline [12]. Among such quasi-experimental studies, there are the well-known socio-psychological experiments of S. Milgram, F. Zimbardo, M. Sheriff, S. Asch, and others [23], as well as the famous experi-

ments of A. Tversky and D. Kahneman [18,19]. Although quasi-experiments are subject to concerns regarding internal validity, because it may not be possible to convincingly demonstrate a causal link between the treatment condition and observed outcomes [51], it was found that quasi-experimental studies in small social groups make it possible to model psychological phenomena that can be found in large social groups [10,11,52,53].

To examine the degree of randomness in humans deciding among equivalent alternatives, we have conducted a triple psychological quasi-experiment designed to capture subject preferences on a continuum of equivalent alternatives when either no restrictions or some restrictions on the admissible alternatives were introduced. Our experiments were set on three gender-balanced groups of students, whose average age was 25 yrs, known as an age prone to random decision-making [16].

The particular experimental design was chosen for: (i.) decision task simplicity for the student subjects; (ii.) lower choice set complexity; (iii.) higher preference uncertainty; (iv.) the relative simplicity of data exploration and processing; (v.) the external similarity of the experimental setup to unmotivated movement in an arbitrarily direction requiring analysis of primary directional information for orienteering. The experimental design was developed in stages, as reflected in the sequential experiments explained below.

Every subject was given a notebook with 10 blank pages and asked to put a period at the center of the first page and imagine a circle around the central point that does not go beyond the page limits. Then, the subject was asked to draw a straight line from the central point of the page to the imagined circle in any direction. After drawing the line, subjects were asked to turn over the page (to avoid any cognitive interference of the new line direction with that of the previously drawn line) and repeat the same operations on the new page again, and so on over all 10 pages of the notebook. No instructions were given for the drawing directions of lines on the sequential pages: all lines might be drawn at the same angles, or at the different ones, as the subject liked. Ten angles representing directions of lines drawn by every subject were recorded, and the data collected over the entire group was analyzed using machine learning algorithms (see Section 4).

Experiment 1:

In the first experiment, no direction restriction was imposed; i.e., subjects were allowed to draw lines in any direction of their choice. The recorded line-drawing angles might be $\alpha \in [0°, 360°]$ (see Figure 1a). The group of subjects participating in the first experiment consisted of 289 students.

Experiment 2:

In the second experiment, subjects could draw lines at any angle, excepting those in the first quadrant (see Figure 1b), so that the recorded line-drawing angles might be $\alpha \in [90°, 360°]$ The group of subjects participating in the second experiment consisted of 98 students.

Experiment 3:

In the third experiment, the admissible sector of line-drawing angles was set to $\alpha \in [90°, 270°]$ (see Figure 1c). The group of subjects participating in the second experiment consisted of 49 students.

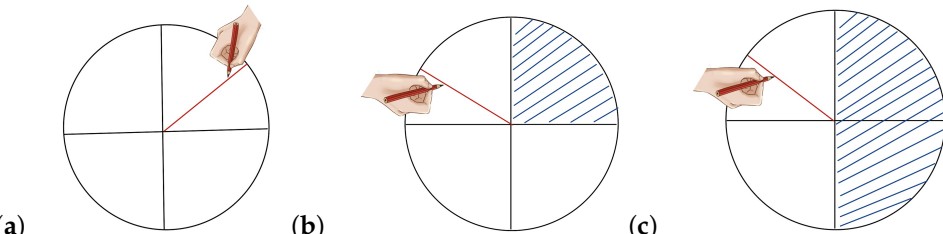

**Figure 1.** The scheme of the triple quasi-experiment on a continuum of equivalent alternatives. (**a**) Experiment 1: subjects were allowed to draw lines in any direction. (**b**) Experiment 2: subjects could draw lines at any angle, excepting those in the first quadrant. (**c**) Experiment 3: the first and second quadrants were barred, but subjects could draw lines at any angle within the left side of circle.

When conducting psychology experiments taking place in a laboratory, a field site, or outside the lab, the subjects may be influenced by a large number of various factors that cannot be completely eliminated. The individual characteristics of people vary, so the problem of individual differences in reaction is considered one of the most important and complex in psychology. Theoretically speaking, we cannot exclude the influence of some factors, such as that being in a university auditorium and receiving a notebook could be associated with taking an exam.

However, in our experiment, there was no requirement to complete a task in limited time. According to the experimental setting, the subjects were not required to demonstrate any intellectual or other capabilities to be compared with their peers. The instructions given to students before the experiment explained that the proposed task had no right or wrong answers, the drawing directions were the matter of their personal choice, which would neither be evaluated by the experimenter, nor by other subjects, nor by third parties, in any way. The participants were also notified that their work would not be compared with the results of other subjects, and all the information obtained in the study would be used only for anonymized computer analysis. No time limits were set in the experiment. The experiment began after *all* subjects confirmed that they understood the instructions and were ready to complete the task. The experiment ended when *all* subjects completed the task proposed to them. The experiment was carried out in small groups in comfortable, well-lit classrooms. The detailed plan of the study was published and widely announced among the students of the Institute of Psychology well before enrollment.

The study did not involve randomization and group classification of participants in the context of intervention under assessment. Each and every participant freely and deliberately read, agreed, and signed the participation consent form. Therefore, we have no reason to believe that the students participating in the experiment were in conditions of anxiety or depression, as they showed a personal interest in taking part in a psychological experiment, familiarized themselves with the experimental setting, and agreed to participate in the study.

In regard to the nature and structure of the experimental data collected, it is worth mentioning that rather than using statistical methods consecrated by tradition, in our work, we implemented *machine learning* methods (explained in the forthcoming section in detail) to find generalizable *predictive* patterns and clusters in the data. While statistics draws population inferences from a sample through the fitting of some hypothesized probability distribution to the data, machine learning makes minimal assumptions about the data-generating systems, and its general-purpose learning algorithms retain their high efficiency even when the data are gathered without a carefully controlled experimental design and in the presence of strong nonlinear interactions [54]. The high individual sensation differences and excessive variability in the personal sensitivity traits of the subjects that participated in our experiment made the use of standard statistical methods ineffective. Initial attempts at usual statistical data processing undertaken immediately after the experiments in 2016 were not successful. However, the use of machine learning methods made it possible to classify the collected data.

## 4. Methods: Machine Learning Algorithms for Data Clustering

The key concept of machine learning is the manifold hypothesis, suggesting that natural data lie along low-dimensional manifolds in high-dimensional data embedding space [55–57]. Algorithms of machine learning (based on some manifold related metric) are designed in a way to separate tangled data manifolds representing meaningful clusters in the data.

### 4.1. Data Embedding and Related Riemann Metric

The data collected in the experiments discussed in Section 3 consist of 10 consecutive angles $(\alpha_{i,1} \ldots \alpha_{i,10})$ representing a sequence of choices made by the $i$-th subject on a continuum of equivalent alternatives. These sequences are rather short to treat as chunks of time series in hopes of finding some common behavior patterns using a statistical approach. To keep a possible sequentiality of angles selected in their temporal order, we embed the 10-angles data onto a 5-dimensional unit sphere, $S_5$, on which every surface point $i$ is characterized by exactly $\begin{pmatrix} 5 \\ 2 \end{pmatrix} = 10$ independent angular coordinates describing elemental rotations $R_{k,j}(\alpha_i)$ in 10 planes of 5-dimensional geometry (e.g., [58]), viz.,

$$R_i = R_{1,2}(\alpha_1)R_{1,3}(\alpha_2)R_{1,4}(\alpha_3)R_{1,5}(\alpha_4)R_{2,3}(\alpha_5)R_{2,4}(\alpha_6)R_{2,5}(\alpha_7)R_{3,4}(\alpha_8)R_{3,5}(\alpha_9)R_{4,5}(\alpha_{10}) \tag{1}$$

The product of elemental rotation matrices (1) describes a rotation in 5-dimensional space in the same way as yaw, pitch, and roll angles, $\nu$, $\beta$, and $\gamma$, describe a rotation in 3-dimensional space; $R_3(\nu, \beta, \gamma) = R_z(\nu)R_y(\beta)R_x(\gamma)$ where $R_{x,y,z}$ are the well-known basic rotation matrices about axes $z, y, x$, respectively. For example, the elemental rotation matrix $R_{2,5}(\alpha_7)$ describing the rotation in the plane $(2,5)$ through the angle $\alpha_7$ is given by

$$R_{2,5}(\alpha_7) = \begin{pmatrix} 1 & 0 & 0 & 0 & 0 \\ 0 & \cos(\alpha_7) & 0 & 0 & -\sin(\alpha_7) \\ 0 & 0 & 1 & 0 & 0 \\ 0 & 0 & 0 & 1 & 0 \\ 0 & \sin(\alpha_7) & 0 & 0 & \cos(\alpha_7) \end{pmatrix}. \tag{2}$$

Given $R_{P1}$ and $R_{P2}$, two rotation matrices representing the sequences of 10 choices made by two participants, $P1$ and $P2$, we find the best approximation for a single rotation, $\Omega_{P1,P2} \in SO(10)$, connecting $R_{P1}$ and $R_{P2}$ (see Figure 2a) to minimize the Frobenius norm of a possible discrepancy, viz.,

$$R_{P1,P2}(\theta) = \min_{\Omega \in SO(5)} \|R_{P1}\Omega_{P1,P2} - R_{P2}\|_F. \tag{3}$$

The minimization step is required in (3) to measure the distance along the *shortest* arc connecting $P1$ and $P2$ (as there are obviously infinitely many arcs connecting to points on $S_5$). Finding the minimum over all matrices of the special orthogonal group $SO(5)$ in (3) is nothing else but the orthogonal Procrustes problem [59], which can be solved by the singular value decomposition of the matrix $R_{P1}^{\top}R_{P2}$, i.e.,

$$\Omega_{P1,P2} = UV^{\top} \quad \text{where} \quad U, V^{\top} : \ R_{P1}^{\top}R_{P2} = U\Sigma V^{\top} \tag{4}$$

where $U^{\top}U = UU^{\top} = I$, $V^{\top}V = VV^{\top} = I$ are the real orthogonal matrices, $I$ represents the identity matrix of appropriate size, and $\Sigma$ is a diagonal matrix of singular values. Obviously, $\Omega_{P1,P2} = I$ if and only if $R_{P1} = R_{P2}$.

Following [60], we define a distance between the sequences of choices made by two participants, $P1$ and $P2$, as a Riemann metric on $S_5$, i.e., as the length of the shortest arc connecting $P1$ and $P2$ along the minimal rotation $\Omega_{P1,P2}$ (see Figure 2), viz.,

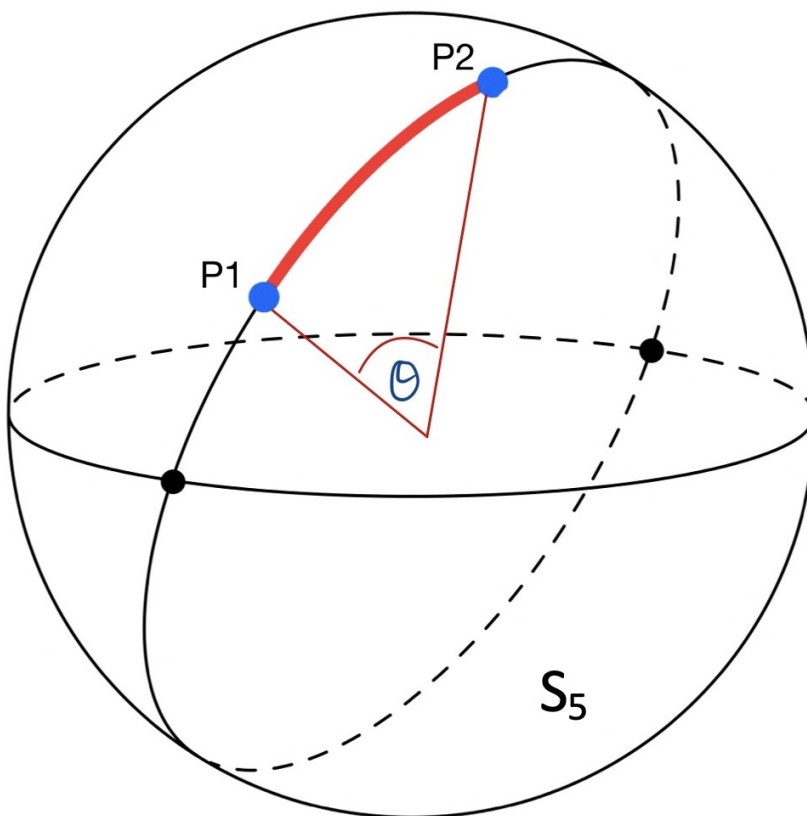

**Figure 2.** The Riemann distance between two participants, $P1$ and $P2$, on $S_5$.

$$D_{P1,P2} \;=\; \|\log \Omega_{P1,P2}\|_F = |\theta_{P1,P2}| \tag{5}$$

It follows from the definition that the Riemann metric defined in (5) satisfies the conditions of non-negativity, identity of indiscernibles, symmetry, and subadditivity. The triangle inequality is satisfied as well, as the length of the geodesic curve on the unit sphere $\exp(t \times \log \Omega_{P1,P2})$ is a strictly positive function for $0 \leq t \leq 1$ [60].

Perhaps a sphere $S_5$ is not the unique candidate for a data manifold relevant to our experiments, as any compact manifold characterized by 10 angular variables might fit the goal, e.g., the torus $S_2 \times S_3$ or more topologically complex compact manifolds with more wholes might be other choices. However, $S_5$ is the topologically simplest manifold and we do not have any reason to relate our experimental settings to manifolds of more complex topology. The effectiveness of our approach is confirmed ultimately by the suitability of results obtained, as the data clusters identified w.r.t. the Riemann metric allow for a meaningful interpretation while the data analysis methods based on the hypothesis that the data are embedded in Euclidean space (such as classical multidimensional scaling) do not allow for reliable clustering. Indeed, there is no reason to believe that human choices expressed by a sequence of lines drawn presumably at random on the different pages of a notebook form a vector in Euclidean space.

### 4.2. Mapping the Data by Stochastic Neighbor Embedding

Clustering is a "many-to-one" mapping from the (high-dimensional) data manifold to a low-dimensional, in our case, 2-dimensional, Euclidean space (of a journal page). One algorithm suitable for such a goal is Stochastic Neighbor Embedding (SNE), which allows embedding of participants according to their preferences in a low-dimensional space so as to optimally preserve statistical neighborhood identity in the sense of data-point distribution [61]. In SNE, the 2-dimensional vectors $(x_i, y_i)$ representing the coordinates of

the *i*-th subject in Euclidean space are determined by minimizing a sum of Kullback–Leibler divergences between the probability distribution over the pairs of subjects on $S_5$,

$$p_{ij} = \frac{\exp\left(-D_{ij}^2\right)}{\sum_{k \neq i} \exp\left(-D_{ik}^2\right)} \tag{6}$$

where the distance $D_{ij}$ is defined in (5) and the distribution in the Euclidean plane is assumed to be Gaussian one, with a fixed variance $1/2$, viz.,

$$q_{ij} = \frac{\exp\left(-(x_i - x_j)^2 - (y_i - y_j)^2\right)}{\sum_{k \neq i} \exp\left(-(x_i - x_k)^2 - (y_i - y_k)^2\right)}. \tag{7}$$

The coordinates of points representing subjects on the plane are chosen in such a way that the following sum of Kullback–Leibler divergences, $K = \sum_{i,j} p_{ij} \log \frac{p_{ij}}{q_{ij}}$, is minimal; i.e.,

$$\frac{\partial K}{\partial x_i} = 2 \sum_j W_{i,j}(x_i - x_j) = 0, \quad \frac{\partial K}{\partial y_i} = 2 \sum_j W_{i,j}(y_i - y_j) = 0 \tag{8}$$

where $W_{i,j} \equiv (p_{ij} + p_{ji} - q_{ij} - q_{ji})$. The gradient Equation (8) show that subjects demonstrating similar choice in the experiments appear to be closer on the plane, and vice versa.

### 4.3. Soft Clustering by Gaussian Mixture Model

In the framework of GMM, data points within a specific cluster are assumed to be generated from a mixture of a combination of $k$ (determined at the previous step described in Section 4.2) multivariate Gaussian distribution components $\mathcal{N}(\mu_s, \Sigma_s)$ with the certain weight $w_s$, mean $\mu_s$, and a covariance matrix $\Sigma_s$ [62]. The mixture is defined by a vector of weights, where each weight $w_s$ represents the fraction of subjects described by a corresponding component [63]. The GMM is fitted to the dataset in the course of the iterative algorithm [64] by maximizing the posterior probability,

$$p(\Theta_1, \ldots \Theta_k | x_i, y_i) = \sum_{s=1}^{k} w_s \mathcal{N}(\mu_s, \Sigma_s) \tag{9}$$

that a data point $(x_i, y_i)$ belongs to its assigned cluster, given the parameters $\{\Theta_s\}_{s=1}^{k}$ of distribution of observations associated with the *s*-th cluster updated at every iteration until model parameters converge.

## 5. Results

The distance matrices $D_{ij}^{(1)}$, $D_{ij}^{(2)}$, and $D_{ij}^{(3)}$, representing the pairwise dissimilarities in subjects' behavior observed in three experiments (Section 3), were calculated from the sequences of line-drawn angles as described in Section 4.1. The primary mapping of the data onto a 2-dimensional plane was performed using the SNE (Section 4.2); then, soft clustering was carried out by the GMM method (Section 4.3).

After the groups of subjects exhibiting similar behavior were identified, the statistical patterns of choice were analyzed further using the autocorrelation functions [65] and radial histograms, showing how often a particular drawing angle was chosen within each group of subjects. The autocorrelation function is defined as Pearson's correlation coefficient between the sequences of 10 angles observed within a group and their copies shifted by one, viz.,

$$\text{ACF}(|\alpha_k - \alpha_{k+1}|) = \frac{\text{Cov}(\alpha_k, \alpha_{k+1})}{\sqrt{\text{Var}(\alpha_k)\text{Var}(\alpha_{k+1})}} \tag{10}$$

where Cov and Var are the covariance and variance of the 10-angle sequences in the group. The autocorrelation function (10) is used to identify the typical lags $|\alpha_k - \alpha_{k+1}|$ between the sequential drawing angles $\alpha_k$ and $\alpha_{k+1}$, $k = 1, \ldots, 9$ within each group of subjects.

The experimental results of our study are as follows.

*5.1. Experiment 1: Engaging Uncertainty through Behavior Patterning*

The structure of the distance matrix $D_{ij}^{(1)}$ representing behavior dissimilarities between the pairs of participants in Experiment 1 is visualized by a phylogenetic tree shown in Figure 3a constructed with the use of the neighbor-joining method [66]. In the phylogenetic tree, subjects are represented by the end points of tree branches. In the framework of the neighbor-joining method, pairs of subjects exhibiting the maximal degree of similarity in their behavior appear close to each other and are connected by the elemental forks representing these pairs. The algorithm then iterates over the obtained pairs of most similar subjects by connecting the central points of forks until the tree is completely resolved (see Figure 3a). The phylogenetic tree constructed by the neighbor-joining method helps to visualize the fine structure of the subjects' grouping.

The results of GMM soft clustering of the data collected in the first experiment are presented in Figure 3b. In the diagram, subjects are shown by points located on the plane of two features representing the most prominent lineaments of angle sequences chosen at random by the participants of Experiment 1.

All participants can be unevenly classified according to their random choice strategies into four groups, as shown in Figure 3b. Subjects belonging to the same clusters exhibited a similar strategy of random choice in Experiment 1, and vice versa. Four groups of subjects were also identified, highlighted by ellipses on the phylogenetic tree shown in Figure 3a.

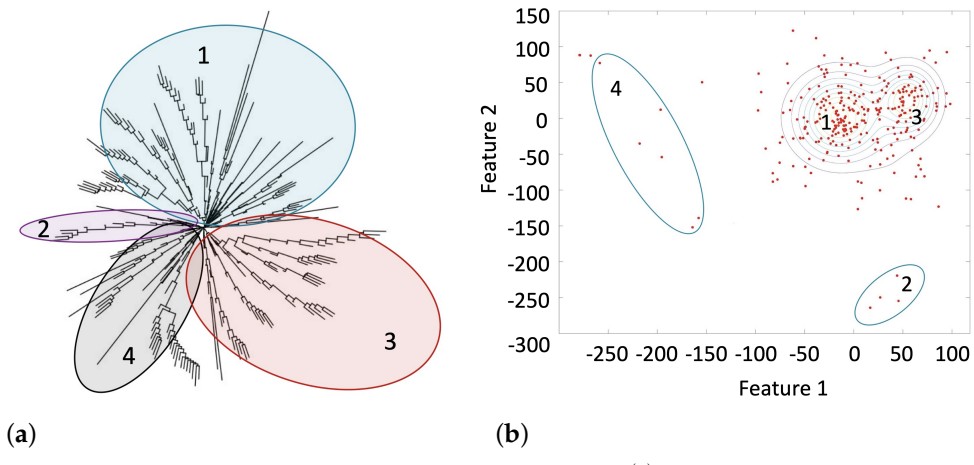

(**a**)                                                                                                    (**b**)

**Figure 3.** (**a**) The fine structure of the distance matrix $D_{ij}^{(1)}$ is visualized by a phylogenetic tree constructed using the neighbor-joining method. The ellipses mark the four groups of subjects following different random choice strategies. (**b**) The results of GMM clustering in Experiment 1 shows four groups of subjects. Features 1 and 2, the coordinates obtained for the subjects when the GMM iteration process converged, are measured in procedure-defined units.

The majority of participants belong to the densely populated Groups 1 and 3, demonstrating strongly similar behavior, while some of them are loosely aggregated into the Groups 2 and 4 (Figure 3a,b). Observed behavior differences are not related to gender. Men and women were equally presented in each group. In Figure 4a, we show a scatter plot representing the male and female participants in the feature space of the experimental data collected in the first experiment by black and white circles, respectively. In Figure 4b, we juxtapose two color-coded kernel density plots of the preferred line-drawing angles observed in Experiment 1 for the male and female subjects. Two humps clearly visible

in Figure 4b for both male and female participants indicate the common, sex-irrelevant preference to draw lines in the upper-right and lower-left quadrants.

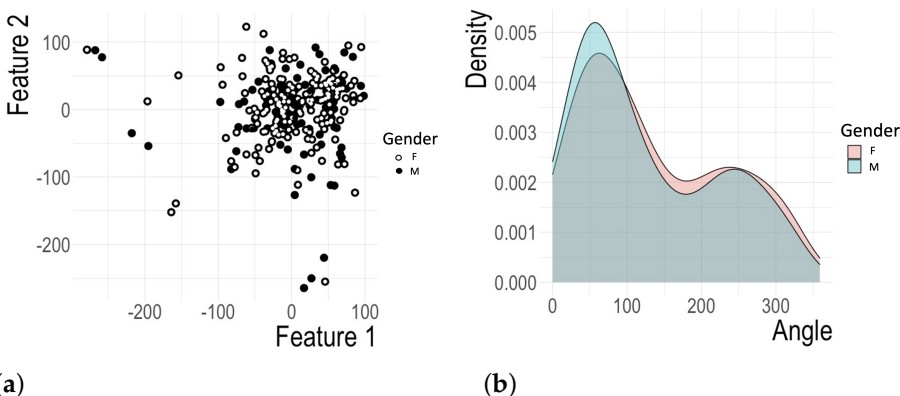

(**a**) (**b**)

**Figure 4.** (**a**) A scatter plot representing male and female participants in the feature space of experimental data collected in Experiment 1. (**b**) The kernel density plots of the line-drawing angles in Experiment 1 for male and female subjects. The Mann–Whitney U-test of the distributions of line-drawing angles shows that there is no statistically significant sex-specific difference in line angel preferences between men and women.

The *p*-value calculated from the Mann–Whitney $U$-test used to compare the sex specific samples shown in Figure 4b is $p = 0.1474 > 0.05$, meaning that there is *no* statistically significant difference between preferred angles of lines drawn by men and women.

The random choice strategies in the four discovered groups are discussed in further detail below.

Group 1, Experiment 1:

The autocorrelation function (10) of line-drawing angles calculated in Group 1 is shown in Figure 5a. The function exhibits sharp peaks at the angle lags multiple of 45°. The dashed horizontal lines in Figure 5a show the 95% confidence interval, indicating the presence of a statistically significant behavior pattern. The first peak at 0°, exceeding the confidence threshold, indicates that in a statistically significant number of cases, the angle lag might be zero, so the drawing angles may be sequentially repeated on the next page. However, in most cases, the next page line is drawn at an angle multiple of 45° against the line drawn on the previous page. The statistically significant lags between the sequential angles are visible at 45°, 90°, 135°, 180°, 225°, and 270°.

In Figure 5b, we show a radial histogram (not normalized) giving an approximate representation of the distribution of angles drawn in a particular direction observed in Group 1. The majority of drawing angles were aligned along the compass directional axes, 45° against each other, i.e., 90°–270°, 0°–180°, 45°–225°, and 135°–315°. Angles in the first quadrant (0°, 45°, and 90°) were chosen more often than angles in other quadrants.

We can conclude this subsection with a remark that, while deciding on a continuum of equivalent alternatives at random, the subjects of Group 1 reduced the available choice to eight directions aligned along the symmetric axes, 45° apart from each other. Subjects from the first group engaged uncertainty of choice through patterning their behavior. They either repeated the same line as was drawn on the previous page of the notebook or chose the direction of a new line randomly from eight major (compass) directions, although three line directions belonging to the first quadrant were more preferable than others.

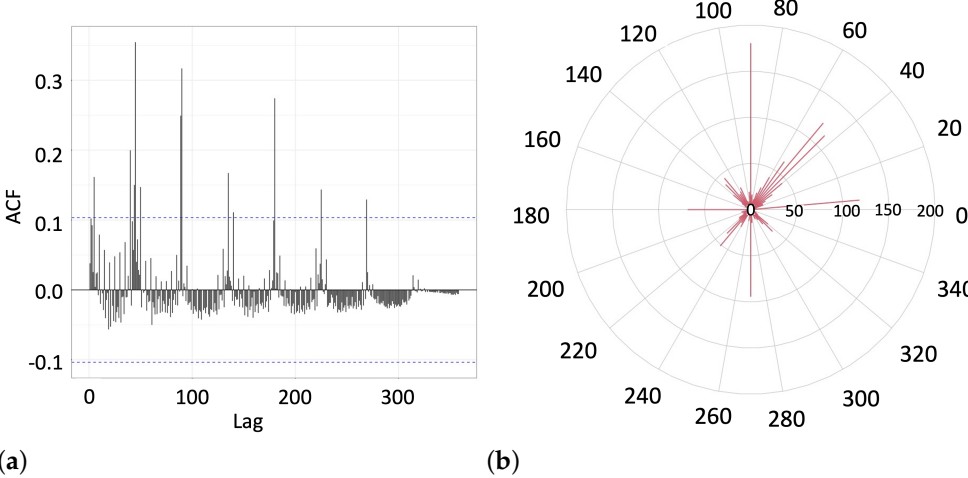

(**a**)                                                                      (**b**)

**Figure 5.** (**a**) The autocorrelation function (10) of the angle lags between the sequential lines observed in Group 1 shows peaks at the lag multiple of 45°. The dashed horizontal lines mark the level of statistically significant correlations. (**b**) The radial histogram (not normalized) represents the approximation for a kernel density estimation of the line angles drawn in particular directions in Group 1.

Group 2, Experiment 1:

The sparse Group 2 (see Figure 3b) featured a single statistically significant peak of the autocorrelation function (10) at the angular lag of 180° against the direction of the previous line (see Figure 6a). Subjects classified into this group simply reversed the direction of the line when turning the notebook page. The radial histogram shown in Figure 6b indicates that the majority of lines drawn by the subjects of Group 2 were along the horizontal axis (0°–180°), with occasional flips to the vertical direction.

While deciding on a continuum of equivalent alternatives at random, subjects from the second group patterned their behavior mostly by flipping the line-drawing direction along the horizontal axes, although the angle of 270° was occasionally used as well.

Group 3, Experiment 1:

The participants classified into Group 3 used a similar direction choice strategy as those aggregated into Group 1. Namely, they either repeated the line drawn on the previous page (the statistically significant peak at a zero lag is presented in Figure 7a), or drew a new line at an angle multiple of 45° against the previous line. Interestingly, the comb of statistically significant peaks of the autocorrelation function (10) presented in Figure 7a shows a tendency to descend with the lag; shorter lags between the sequential angle multiple of 45° were chosen more often than longer lags.

The radial histogram presented in Figure 7b shows that the majority of lines were aligned along the compass directional axes, at 45° to each other. However, in contrast to the choice strategy observed in Group 1, the line directions were distributed more evenly in the second, third, and fourth quadrants of the plane, although line directions belonging to the first quadrant were still more preferable than others. We may suggest that participants aggregated into Group 3 filled out their notebooks by drawing lines along the major (compass) directional axes in a clockwise (or counterclockwise) direction. While subjects of the first group showed a clear preference for the first quadrant in line-drawing directions, Group 3 tended to cover other quadrants more evenly, perhaps due to implementing a clockwise/counterclockwise line-drawing strategy.

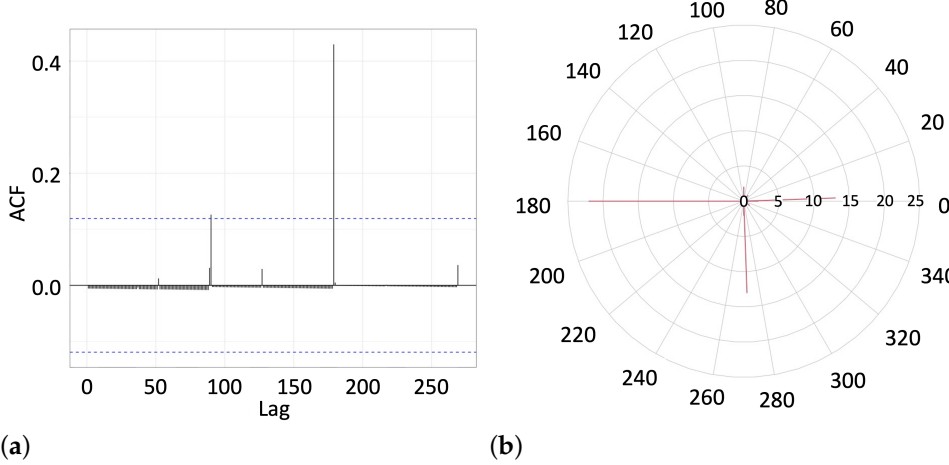

(a)                                                                  (b)

**Figure 6.** (**a**) The autocorrelation function (10) represents a single statistically significant lag (of 180°) between the sequentially drawn lines in Group 2. The dashed horizontal line indicates the level of statistically significant correlations. (**b**) The radial histogram (not normalized) of the line-drawing angles observed in Group 2.

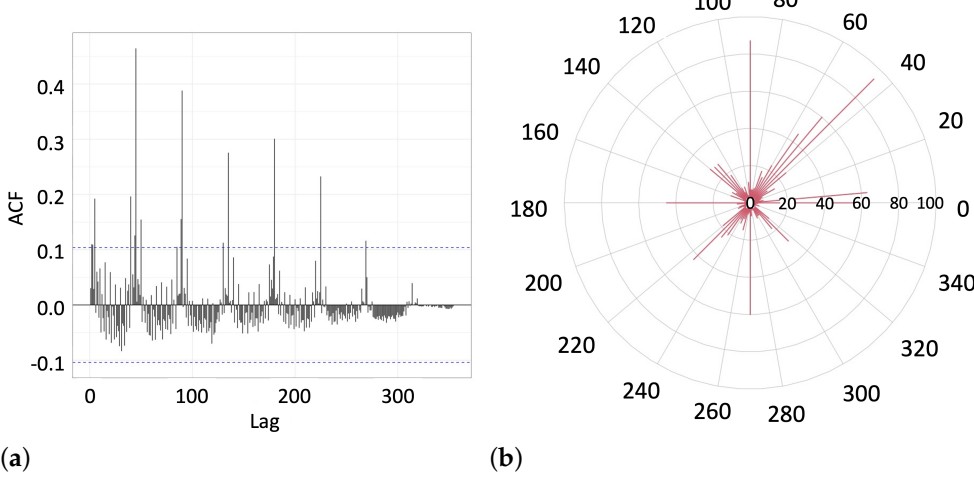

(a)                                                                  (b)

**Figure 7.** (**a**) The autocorrelation function (10) represents the statistical significance of lags between the sequential line angles in Group 3. The dashed horizontal lines indicate the level of statistically significant correlations. (**b**) The radial histogram (not normalized) of the line-drawing angles observed in Group 3.

Group 4, Experiment 1:

The autocorrelation function (10) of lags between the sequential angles of the participants aggregated into Group 4 is presented in Figure 8a. The graph shows a single marginally significant peak at 90°. A line drawn on the previous page was never repeated on the next page. The radial histogram presented in Figure 8b clarifies that subjects of Group 4 mostly flipped two line directions (0° and 90°) at random, in a manner of coin tossing (as the angle lag of 90° is only marginally significant, as follows from Figure 8a). The line angles of 180° and 270° were observed occasionally.

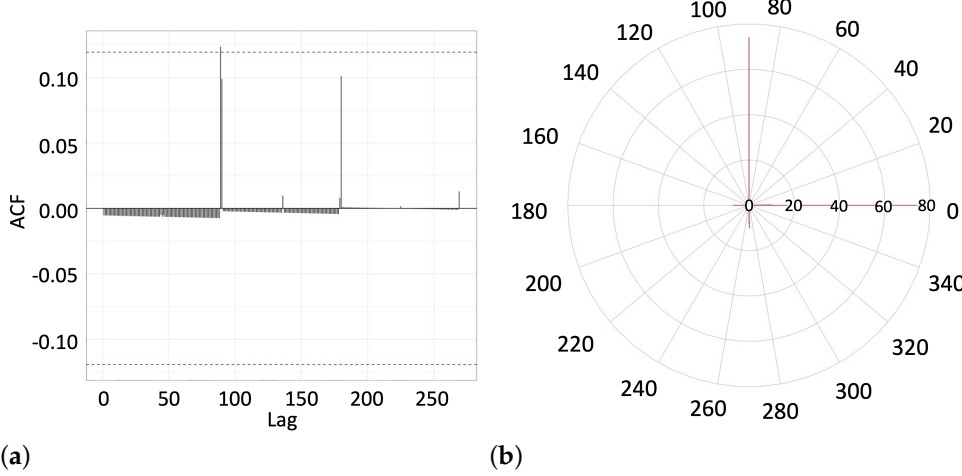

(**a**)                                      (**b**)

**Figure 8.** (**a**) The autocorrelation function (10) represents the statistical significance of lags between the sequential line angles in Group 4. The dashed horizontal line marks the level of statistically significant correlations. (**b**) The radial histogram (not normalized) of the line-drawing angles observed in Group 4.

*5.2. Experiment 2: Breaking Behavior Patterns through Imposed Restrictions*

In the second experiment, the first quadrant was barred from drawing lines, and therefore many behavior patterns observed in Experiment 1 (Section 5.1) were unfeasible. The structure of the distance matrix $D_{ij}^{(2)}$ is visualized by the phylogenetic tree shown in Figure 9a, constructed with the use of the neighbor-joining method as discussed previously. The GMM soft clustering method applied to the data collected in Experiment 2 aggregates all subjects into a single cluster presented in Figure 9b.

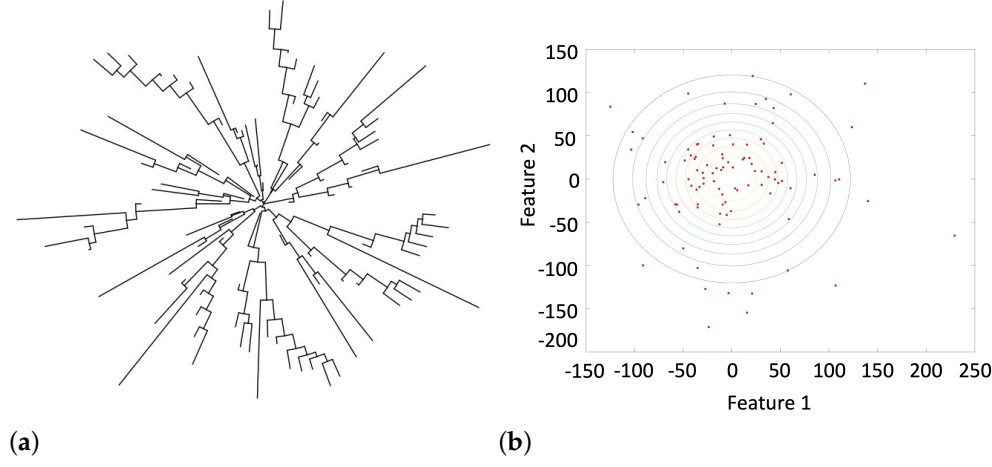

(**a**)                                      (**b**)

**Figure 9.** (**a**) The structure of the distance matrix $D_{ij}^{(2)}$ visualized by a phylogenetic tree with the use of the neighbor-joining method. (**b**) The results of GMM clustering aggregates all participants of Experiment 2 into a single cluster. Features 1 and 2 (the coordinates obtained for subjects when the iteration process converged) are measured in procedure-defined units.

The autocorrelation plot shown in Figure 10a gives evidence that subjects did not repeat the previously drawn lines on the next page of the notebook. The statistically significant angle lags were multiples of $45°$, although the shares of lines drawn at the intermediate angles increased substantially, as shown in the radial histogram in Figure 10b.

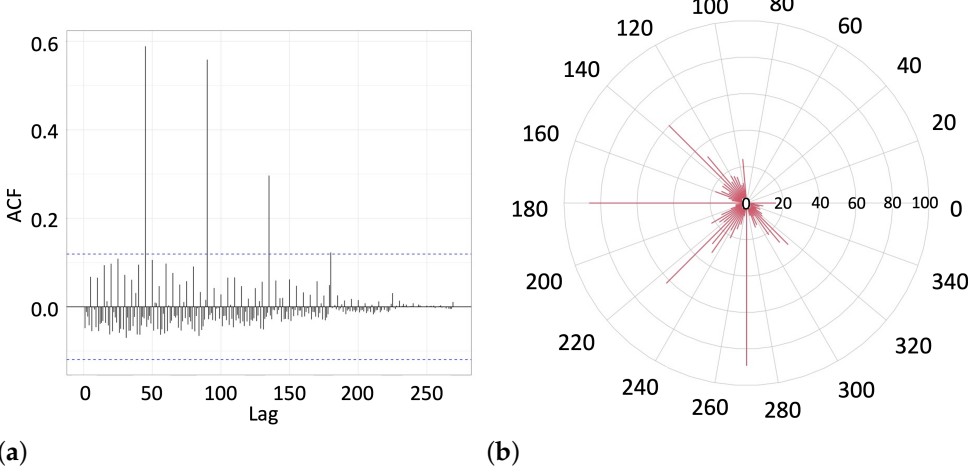

**(a)** **(b)**

**Figure 10.** (**a**) The autocorrelation function (10) of the angle lags between the sequential lines observed for the participants of Experiment 2. The dashed horizontal line marks the level of statistically significant correlations. (**b**) The radial histogram (not normalized) of the line-drawing angles observed in Experiment 2.

The restrictions imposed in the second experiment barred the use of the first quadrant, which was the most popular choice among the participants of the first experiment. Although the angular lag multiple of 45° was still a statistically significant line direction choice in the second experiment, the participants drew lines not only along the major axes, but also at the intermediate angles, apparently trying to cover the available angular sector more evenly.

*5.3. Experiment 3: No Common Choice Preferences When More Restrictions Imposed*

Finally, the first and second quadrants were barred from use in the third experiment. While making a choice under the imposed constraints, subjects showed a tendency to use the available half-circle more evenly by drawing lines at random directions centered at 180°, bisecting the admissible sector (Figure 11b).

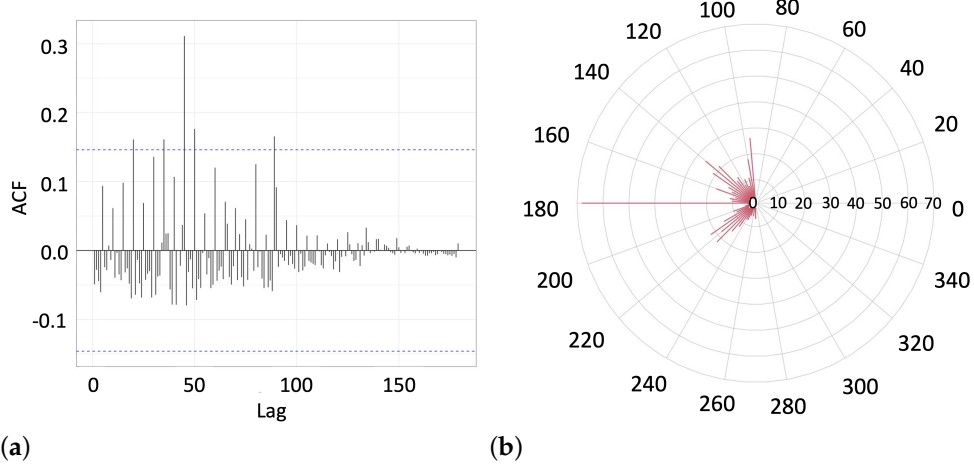

**(a)** **(b)**

**Figure 11.** (**a**) The autocorrelation function (10) of the angle lags between the sequential lines observed in the group of participants in Experiment 3. The dashed horizontal line marks the level of statistically significant correlations. (**b**) The radial histogram (not normalized) of the line-drawing angles observed in the third experiment.

The autocorrelation plot of the data collected in the third experiment (Figure 11a) shows a few marginally significant peaks at different angle lags, including the 45°-lag ubiquitous in our experiments. The radial histogram presented in Figure 11b shows that

the line-drawing angles cover almost the entire admissible sector, although the bisector line at 180° was used most often.

We investigated the statistics of line-drawing angles in the group of participants in the third experiment. In Figure 12a, we show the empirical probability density plot of drawn angles observed in the third experiment, with a reference line representing the normal distribution $\mathcal{N}(180.29°, 8.18)$ best fitted to the data. To compare the empirical distribution of drawn angles to the normal distribution $\mathcal{N}(180.29°, 8.18)$, we also presented a quantile–quantile normal probability plot [67] in Figure 12b.

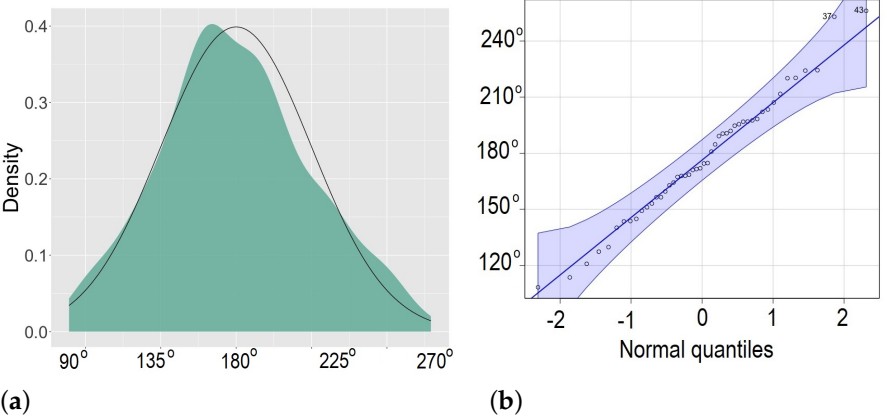

(**a**)             (**b**)

**Figure 12.** (**a**) The probability density plot of the line-drawing angles in Experiment 3, with a reference line representing the normal distribution $\mathcal{N}(180.29°, 8.18)$ fitted the data best. (**b**) The quantile–quantile normal probability plot shows quantiles of the empirical drawing angle distribution against quantiles of the normal distribution $\mathcal{N}(180.29°, 8.18)$. The similarity of both distributions are justified by the fact that points are almost perfectly aligned along the diagonal line $y = x$. The shaded area defines the 95% confidence bounds for the normal distribution quantiles, with mean and standard deviation calculated from data (angles).

In a normal probability plot (Figure 12b), the collected angular data were plotted against values sampled from the normal distribution $\mathcal{N}(180.29°, 8.18)$ to make the resulting curve appear close to a straight line $x = y$, as the data are approximately normally distributed. Deviations from a straight line suggest departures from normality. The empirical line looks fairly straight, at least when the few large and small values are ignored, indicating that the empirical distribution of line-drawing angles was close to normal. Indeed, an empirical distribution close to normal (Figure 12b) appears in the data collected from the group of participants. Individual strategies of coping with uncertainty of choice under imposed restrictions may be very diverse.

## 6. Conclusions and Discussion

In our work, we reported the results of psychology experiments in which the gender-balanced groups of participants were asked to draw 10 straight lines randomly directed from the center of a page on 10 pages of a blank notebook.

Our experiments demonstrated that while facing a dilemma of choice on a continuum of equivalent alternatives (line-drawing directions), subjects tend to engage uncertainty through patterning their behavior. They drew lines along the main compass axes in a circle (Figure 13), although their individual strategies greatly varied, ranging from sequential filling-in of the notebook pages with lines drawn at 45° against each other in a clockwise direction, to a random flipping of the previous line in a horizontal (or vertical) direction. The common behavior pattern represented in Figure 13 may appear due to a standardized schooling experience or the common practice of using a dial in ubiquitous technical devices, such as clocks, measures, and compasses.

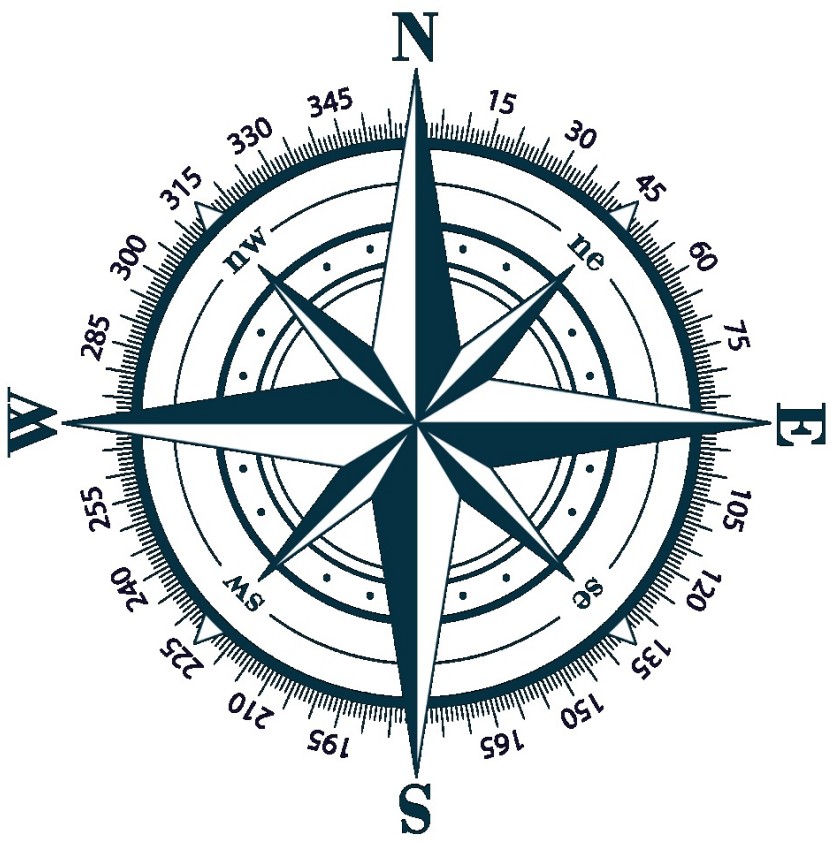

**Figure 13.** The main compass axes featured by the line-drawing strategies on a continuum of equivalent alternatives.

Under the restrictions imposed in the second and third experiments, some drawing directions of the main axes (Figure 13) appeared unfeasible to use; then, subjects drew lines in the admissible sector more evenly, following individual strategies rather than a common pattern. The analysis of data collected from the group of participants shows that the observed distribution of line-drawing angles was fairly close to a normal one centered at the horizontal direction 180°, bisecting the admissible sector. Approximately normal distributions occur in many situations, especially when the outcome is produced by many small effects acting additively and independently of each other [68].

The appearance of a normal distribution in the experimental data collected from a group of participants shows a lack of common strategy to fill the available sector rather than true randomness in the choice of line-drawing direction.

Our results support the general conclusion of previous studies [8–12] that *abundance of subjectively equivalent alternatives may reduce the individual variability of choices* made by humans seeking to eliminate uncertainty. Furthermore, we demonstrated that this relationship might be also true the other way around; *we may achieve increasing variability in human decisions by restricting the range of available equivalent alternatives to choose from.*

As a rational choice under uncertainty is impossible, subjectively equivalent options abound, which is usually associated with the idea of freedom and autonomy, potentially increasing the risk of unknown outcomes and missed opportunities. In particular, a dramatic explosion in choice in equivalent goods may become an unbearable burden for customers, while eliminating consumer choices can greatly reduce anxiety for shoppers [69]. Moreover, greater levels of choice overload are associated with greater probability of choice deferral, greater switching likelihood, decreased preference for larger assortments, and greater preference for easily justifiable options [70]. Not surprisingly, when confronting uncertainty of choice, people try to *evade personal decision-making* at all costs, often with the use of a variety of impersonal methods of choosing, such as coin tossing [17]. Under

such circumstances, *defaults and guidelines* that would reduce freedom of choice, but lessen the number of decisions that people are forced to make on their own would reduce the psychological stress that people may face, and therefore improve their well-being [17].

We believe that our observation might provide a key for understanding the surprising effectiveness of modern AI algorithms in influencing human decision-making under *chronic choice overload* conditions. Perhaps the accuracy of determining the individual traits and inferring the behavior of an internet user in a state of choice overload is not that important, as people may appreciate *any* supervision to help them to avoid personally parsing the vast array of choices they confront. It is also possible that a situation similar to choice overload can take place in political governance as well. In particular, although it is relatively easy to hold elections, democratic institutions might not function efficiently under conditions of uncertainty [71], as people may have a powerful need to see authority as both strong and benevolent enough to eliminate countless troublesome choices in one's daily life, even in the face of evidence to the contrary [72].

Further research is needed to focus on possible mechanisms of appealing to chance and guidance under uncertainty.

**Author Contributions:** Conceptualization, A.L. and D.V.; methodology, A.L., D.V. and K.R.; software, K.R.; validation, K.R., D.V. and A.L.; formal analysis, K.R. and D.V.; investigation, A.L. and K.R.; resources, A.L. and D.V.; data curation, A.L. and K.R.; writing—original draft preparation, K.R. and D.V.; writing—review and editing, D.V. and A.L.; visualization, K.R.; supervision, D.V.; project administration, A.L. and D.V.; funding acquisition, D.V. All authors have read and agreed to the published version of the manuscript.

**Funding:** Our study was carried out as part of Research Project No. 16-06-00761a, "Rational and non-rational decisions in conditions of equal choice in marketing", supported by the Russian Foundation for the Humanities in 2016–2018, at the Institute of Psychology of the Russian Academy of Sciences.

**Institutional Review Board Statement:** The study was conducted in accordance with the Declaration of Helsinki. The plan of the study and the protocol of experiments were reviewed by the Institutional Review Board of the Institute of Psychology of the Russian Academy of Sciences and approved as a non-interventional and observational study in May 2016 in the framework of Research Project No. 16-06-00761a, "Rational and non-rational decisions in conditions of equal choice in marketing", supported by the Russian Foundation for the Humanities.

**Informed Consent Statement:** Informed consent was obtained from all subjects involved in the study. The study protocol was published and widely announced among the student participants in the Institute of Psychology of the Russian Academy of Sciences before enrollment. Written informed consent was obtained from all participants to publish this paper.

**Data Availability Statement:** Not applicable.

**Acknowledgments:** The authors are grateful to their institutions for the administrative and technical support, as well as to all student subjects who participated in the experiments. A.L. acknowledges the support from the Russian Foundation for the Humanities.

**Conflicts of Interest:** The authors declare no conflict of interest.

## Abbreviations

The following abbreviations are used in this manuscript:

| | |
|---|---|
| AI | Artificial intelligence |
| SNE | Stochastic Neighbor Embedding |
| GMM | Gaussian Mixture Modeling |

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
