# Peer review of "Deciding on a Continuum of Equivalent Alternatives Engaging Uncertainty through Behavior Patterning"

_foundations, doi:10.3390/foundations2040071_

Round 1

Reviewer 1 Report

In this paper, the authors study a psychological experiment on deciding over a continuum of equivalent alternatives. Moreover, the authors investigate the experiment in the restricted settings.

The results of this paper are interesting. The method based on machine learning algorithms is also interesting and well presented. The paper is well organized. On the other hand, the quality of presentation could be improved.

Author Response

Response to Reviewers

We profoundly thank our reviewers for their kind suggestions and clarifying questions inspiring us to revise our manuscript thoroughly. To mention just a few of major modifications made to the text, we rewrote the paper abstract, added a new Introduction section (Sec.1) defining a broad context of our work, added a justification for the experimental design (in Sec.3), added an extensive discussion on screening anxiety and depression in subjects, as well as on the personality traits (in Sec.3). In Sec.5, we added a special paragraph and two new figures demonstrating a lack of statistically significant behavioral differences between male and female subjects in our experiments. In Sec.6, we added an extensive discussion on our results in the broad context. Some (13) new references have been added to the list in the course of paper revision. Finally, the required information on Institutional Review Board Statement, Funding, and Informed Consent Statements have been added to the supplementary materials.

We help our reviewers for the invaluable help again!

Our detained responses to the reviewer’s comments are given below:    

Response to Reviewer 1

The quality of presentation could be improved

We profoundly thank our reviewer! We have revised the manuscript thoroughly and hope that the actual quality of presentation is much better now.

We greatly appreciate time and efforts of our reviewers helping us to improve the quality of our manuscript!

Reviewer 2 Report

This paper focuses on how people make choice when facing multiple choices. In this study, it conducts a series of psychological experiments to measure the behavior patterns. The triple experiment on choosing over a continuum of equivalent alternatives is relatively feasible, and hence it can be repeated to test the outcomes in a wider range. This paper is well written, and contributes to machine learning algorithms scheme for behavior pattern, which enables better processing and analysis of data. Furthermore, the proposed scheme outperforms the status of the arts and can provide a good train of thought for future study.

However, there are some major concerns, which must be solved before it is considered for publication. If the following problems are well-addressed, I believe that the essential contribution of this paper is important for this field. Thus, I think the paper can be considered for publication after major revision.

(1) In Sec. 1, the shortcomings of previous studies need to be more specific and elaborative.

(2) The significance of this paper is not expounded sufficiently. The authors need to highlight the contributions or significance based on the aim and questions in Sec. 1.

(3) In Sec. 2, the design of the experiments cannot convince me. So, I suggest that the authors should state the reasons why you implement the experiments in this way. Further explanation is needed.

(4) Sec. 5 needs more discussion as it is too brief to fully highlight the findings and afterthought of this work.

Author Response

Response to Reviewers

We profoundly thank our reviewers for their kind suggestions and clarifying questions inspiring us to revise our manuscript thoroughly. To mention just a few of major modifications made to the text, we rewrote the paper abstract, added a new Introduction section (Sec.1) defining a broad context of our work, added a justification for the experimental design (in Sec.3), added an extensive discussion on screening anxiety and depression in subjects, as well as on the personality traits (in Sec.3). In Sec.5, we added a special paragraph and two new figures demonstrating a lack of statistically significant behavioral differences between male and female subjects in our experiments. In Sec.6, we added an extensive discussion on our results in the broad context. Some (13) new references have been added to the list in the course of paper revision. Finally, the required information on Institutional Review Board Statement, Funding, and Informed Consent Statements have been added to the supplementary materials.

We help our reviewers for the invaluable help again!

Our detained responses to the reviewer’s comments are given below:    

Response to Reviewer 2

In Sec. 1, the shortcomings of previous studies need to be more specific and elaborative.

The significance of this paper is not expounded sufficiently. The authors need to highlight the contributions or significance based on the aim and questions in Sec. 1

We profoundly thank our reviewer for the inspiring suggestion! The entire Introduction section (Sec.1) has been written anew. The significance of the paper and relation to previous work are clearly highlighted and concisely explained:
1. Introduction 12

Artificial intelligence (AI) is believed to demonstrate the rapidly growing capabilities to 13

predict human behavior, to identify vulnerabilities in human habits, and to use them to 14

steer human decision-making through manipulative interactions [1]. Countless studies 15

discussing the use of AI for understanding how to work with humans suggest that human 16

choice is anything but random, as being predetermined by personal traits and attributes hid- 17

den in behavioral complexity that can be, nevertheless, discovered by the sophisticated AI 18

algorithms through the automated identification of patterns in digital records of individual 19

behavior. 20

For example, Facebook "likes" were found to accurately predict sensitive personal 21

information, such as sexual orientation, ethnicity, religious and political views, personality 22

traits, intelligence, happiness, addiction, parental separation, age, and gender [2]. The 23

information obtained by tracking user’s liking activity may be used for personalizing 24

custom-picked content to nudge a person toward particular actions. According to the 25

Center for Humane Technology [3], the attention of an entire nation can be purchased 26

for the price of a used car, as around 68% of adults in the US are using Facebook for 27

posting pics, shopping, identifying places where they can dine, creating events and sharing 28

awareness [4]. 29

Indeed, the AI algorithms are great at solving specific problems, but only as long as 30

you stick to the script. My own (D.V.) struggle with Facebook has been going on for many 31

years. My courtesy requires me to like every single post of my friends. My likes are a 32

simple sign of respect for a dear friend and do not express my actual (dis)agreement with 33

posted texts (that I almost never read). Due to my unusual liking strategy, I was regularly 34

misidentified as a bot distracting the deep learning process in the platform recurrent neural networks, and therefore blocked for liking too many friends’ pages. Later they began to set 36

a quota for the number of likes I could use, which I usually filled up in a short amount of 37

time to get blocked again. Then, they tried to hide from me the new posts in my friends 38

feed. In response, I searched for my friends’ pages by name and liked all their posts in a row. 39

After a while, Facebook gave up and started showing me some random ads, accompanied 40

by a question whether they met my interests and needs. Needless to say that I deliberately 41

ignore these questions and diligently block the advertisers. 42

All this would be funny if we were not continuously exposed to a great many of 43

unsolicited manipulative experiments involving a variety of AI algorithms, and many 44

of them can be seen as similar to hypnotic suggestions, for which none of us gave any 45

consent. While some innocuous AI-based algorithms may improve our lifestyle by creating 46

healthier dietary habits, others might massively compromise the democratic procedures 47

through election meddling, hack the national economy by steering the responsible decision 48

makers towards accepting faulty, biased, or malicious policies, and even impel people 49

craving for belonging to join military service in the course of unconstitutional mobilization 50

for fighting in a neighboring country, against which the war has not even been declared. 51

Who should then be held accountable for the war crimes enthusiastically committed 52

by soldiers indoctrinated by irresistibly convincing AI-propaganda? To what extent are 53

we liable for our decisions and actions if manipulated by algorithms? What degree of 54

responsibility for the crisis should be borne by the AI developers and social media platforms 55

that implemented these "weapons of math destruction" [5] worldwide? 56

In modern psychology and especially in neuropsychology, there are ongoing dis- 57

cussions about the extent to which human decision-making under uncertainty may be 58

predicted and inferred [6,7]. Drawing a decisive conclusion about possible psychological 59

mechanisms beyond making a choice would be important for solving many scientific 60

and practical problems, such as the philosophical problem of free will and determinism 61

motivated by concerns about moral responsibility for our personal actions, understanding 62

scanning and searching activities in humans and animals, predicting individual’s invest- 63

ment behavior, etc. Perhaps, the most common type of uncertainty is choosing between two 64

or more subjectively equivalent alternatives illustrated by the famous parable of Buridan’s 65

ass. 66

One may believe that making a choice under uncertainty of many equivalent alter- 67

natives can involve some random, as well as deterministic actions. In many studies, e.g. 68

[812], it was demonstrated that, paradoxically, the more subjectively equivalent alternatives 69

available for a subject, the less random the choice made seems to be (when observed over a 70

group of subjects). Confronting a multitude of subjectively equivalent alternatives, humans 71

may try to reduce uncertainty of choice by following some common choice patterns, indeed. 72

In our work, we support this observation and show that it looks also true the other way 73

around. Namely, by restricting the range of available subjectively equivalent alternatives, 74

we may achieve increasing variability in decisions made over a group of subjects. 75

After a review of results on and discussions on whether humans can be random in 76

Sec. 2, we report on the results of a triple psychological experiment, in which three gender 77

balanced groups of subjects were offered to make a choice over a continuum of subjectively 78

equivalent alternatives (directions) (see Sec. 3 for further information). The analysis of 79

experimental data with the use of machine learning algorithms (described in Sec. 4) shows 80

that while making their choice, subjects followed a common pattern giving a preference to 81

just a few directions (featured by the main compass axes, as discussed in Sec. 6) over all 82

others (Sec. 5) although the individual strategies implemented to fulfill the common pattern 83

may greatly vary (Sec. 5.1). By restricting the experimental settings further (Sec. 3), we 84

broke down the common pattern observed in the first experiment, making subjects to follow 85

individual patterns of choice over a continuum of equivalent alternatives (Sec. 5.2,5.3). 86

The experiments revealed no gender specific differences in the random decision making 87

processes. We conclude in the last section (Sec. 6). 88

In Sec. 2, the design of the experiments cannot convince me. So, I suggest that the authors should state the reasons why you implement the experiments in this way. Further explanation is needed.

We profoundly thank our reviewer! The following justification of the experimental design have been added to the text of Experiment section:

The particular experimental design was chosen for (i.) decision task simplicity for the 208

student subjects, (ii.) lower choice set complexity, (iii.) higher preference uncertainty, (iv.) 209

the relative simplicity of data exploration and processing, (v.) the external similarity of the 210

experimental setup to the unmotivated movement in an arbitrarily direction requiring the 211

analysis of primary directional information for orienteering. The experimental design was 212

developed in stages, as reflected in the sequential number of experiments explained below.

and
When conducting psychology experiments taking place in a laboratory, a field site, 239

or outside the lab, the subjects may be influenced by a large number of various factors 240

that cannot be completely eliminated, indeed. The individual characteristics of people 241

vary, so that the problem of individual differences in reaction is considered one of the 242

most important and complex in psychology. Theoretically speaking, we cannot exclude 243

the influence of some factors, such as being in a university auditorium and receiving a 244

notebook could be associated with taking an exam. 245

However, in our experiment, there was no requirement to complete the task in limited time. According to the experimental setting, the subjects were not required 248

to demonstrate any intellectual or other capabilities to be compared with their peers. The 249

instructions given to students before the experiment explained that the proposed task 250

had no right or wrong answers, the drawing directions are the matter of their personal 251

choice that would neither be evaluated by the experimenter, nor by other subjects, nor by 252

third parties, in any way. The participants were also notified that their work would not be 253

compared with the results of other subjects, and all the information obtained in the study 254

would be used only for the anonymized computer analysis. No time limits were set in the 255

experiment. The experiment began after all subjects confirmed that they understood the 256

instructions and were ready to complete the task. The experiment ended when all subjects 257

completed the task proposed to them. The experiment was carried out in small groups in 258

comfortable, well-lit classrooms. The detailed plan of the study was published and widely 259

announced among the students in the Institute of Psychology well before the enrollment. 260

The study did not involve randomization and group classification of participants in the 261

context of intervention under assessment. Each and every participant freely and deliberately 262

read, agreed, and signed the participation consent form. Therefore, we have no reason to 263

believe that the students participating in the experiment were in conditions of anxiety or 264

depression, as they showed a personal interest in taking part in a psychological experiment, 265

familiarized themselves with the experimental setting, and agreed to participate in the 266

study. 267

In regard to the nature and structure of the experimental data collected, it is worth 268

mentioning that rather than using statistical methods consecrated by tradition, in our work, 269

we implemented machine learning methods (explained in the forthcoming section in detail) 270

to find generalizable predictive patterns and clusters in the data. While statistics draws 271

population inferences from a sample through the fitting of some hypothesized probability 272

distribution to the data, machine learning makes minimal assumptions about the data- 273

generating systems, and its general-purpose learning algorithms retain their high efficiency 274

even when the data are gathered without a carefully controlled experimental design and 275

in the presence of strong nonlinear interactions [54]. The high individual sensation differences and excessive variability in personal sensitivity traits of subjects participated in 277

our experiment made the use of standard statistical methods ineffective. Initial attempts 278

at usual statistical data processing undertaken immediately after the experiments in 2016 were not successful. However, the use of machine learning methods made it possible to 280

classify the collected data.

Sec. 5 needs more discussion as it is too brief to fully highlight the findings and afterthought of this work.

We thank our reviewer for the advice! The following text has been added to the Discussion section:
Our results support the general conclusion of previous studies [812] on that abundance 520

of subjectively equivalent alternatives may reduce the individual variability of choices made by 521

humans seeking to take the edge off uncertainty. Furthermore, we have demonstrated 522

that this relationship might be also true the other way around: we may achieve increasing 523

variability in human decisions by restricting the range of available equivalent alternatives to choose 524

from. 525

As a rational choice under uncertainty is impossible, subjectively equivalent options 526

galore, which is usually associated with the idea of freedom and autonomy, would poten- 527

tially increase the risk of unknown outcomes and missed opportunities. In particular, a 528

dramatic explosion in choice over equivalent goods may become an unbearable burden 529

for customers while eliminating consumer choices can greatly reduce anxiety for shoppers 530

[69]. Moreover, greater levels of choice overload are associated with greater probability of 531

choice deferral, greater switching likelihood, decreased preference for larger assortments, 532

and greater preference for easily justifiable options [70]. Not surprisingly, confronting 533

uncertainty of choice, people try to evade personal decision making at all costs, often with 534

the use of a variety of impersonal methods of choosing, such as coin tossing [71]. Un- 535

der such circumstances, defaults and guidelines that would reduce freedom of choice but 536

lessen the number of decisions that people forced to make on their own would reduce the 537

psychological stress that people may face, and therefore improve their well-being [71]. 538

We believe that our observation might provide a key for understanding the surprising 539

effectiveness of the modern AI algorithms in influencing human decision-making under 540

chronic choice overload conditions. Perhaps, the accuracy of determining individual traits 541

and inferring the behavior of an internet user being in a state of choice overload is not that 542

important, as people may appreciate any supervision would help them to avoid personal 543

parsing the vast array of choices they confront. It is also possible that a situation similar to 544

choice overload can take place in political governance as well. In particular, although it is 545

relatively easy to hold elections, democratic institutions might not functioning efficiently 546

under conditions of uncertainty [72], as people may have a powerful need to see authority 547

as both strong and benevolent enough to eliminate countless troublesome choices in one’s 548

daily life, even in the face of evidence to the contrary [73]. 549

Further research is needed to focus on possible mechanisms of appealing to the chance 550

and guidance under uncertainty

We greatly appreciate time and efforts of our reviewers helping us to improve the quality of our manuscript!

Reviewer 3 Report

This is an interesting design aiming to demonstrate the decision-making among choice-decrease or choice-limitation conditions. The protocol is nice and the data collection process is sometimes promising, but the authors must consider the following comments. Decision-making is a rather complicated process, which might be influenced by a variety of factors, from physiology to psychology at least. Individual differences might be especially a factor to be considered. Therefore,

1. Were the participants screened by anxiety or depression exclusions? The anxiousness, especially that is related to time consideration, influences decision-making, drawing performance, and so on.

2. Regarding the individual difference, personality traits should be considered in the study also. Some participants might be higher sensation seekers, and some might be lower ones. Impulsivity plays a crucial role during the process described in this study.

3. Gender (sex) differences in behavioral patterns might be explored in the current study.

4. Minor, the outlet of the Abstract might be structured, for a more understandable effect.

Author Response

Response to Reviewers

We profoundly thank our reviewers for their kind suggestions and clarifying questions inspiring us to revise our manuscript thoroughly. To mention just a few of major modifications made to the text, we rewrote the paper abstract, added a new Introduction section (Sec.1) defining a broad context of our work, added a justification for the experimental design (in Sec.3), added an extensive discussion on screening anxiety and depression in subjects, as well as on the personality traits (in Sec.3). In Sec.5, we added a special paragraph and two new figures demonstrating a lack of statistically significant behavioral differences between male and female subjects in our experiments. In Sec.6, we added an extensive discussion on our results in the broad context. Some (13) new references have been added to the list in the course of paper revision. Finally, the required information on Institutional Review Board Statement, Funding, and Informed Consent Statements have been added to the supplementary materials.

We help our reviewers for the invaluable help again!

Our detained responses to the reviewer’s comments are given below:    

Response to Reviewer 3

Were the participants screened by anxiety or depression exclusions? The anxiousness, especially that is related to time consideration, influences decision-making, drawing performance, and so on. Regarding the individual difference, personality traits should be considered in the study also. Some participants might be higher sensation seekers, and some might be lower ones. Impulsivity plays a crucial role during the process described in this study.

We profoundly thank our reviewer! The following justification of the experimental design have been added to the text of Experiment section:

The particular experimental design was chosen for (i.) decision task simplicity for the 208

student subjects, (ii.) lower choice set complexity, (iii.) higher preference uncertainty, (iv.) 209

the relative simplicity of data exploration and processing, (v.) the external similarity of the 210

experimental setup to the unmotivated movement in an arbitrarily direction requiring the 211

analysis of primary directional information for orienteering. The experimental design was 212

developed in stages, as reflected in the sequential number of experiments explained below.

and
When conducting psychology experiments taking place in a laboratory, a field site, 239

or outside the lab, the subjects may be influenced by a large number of various factors 240

that cannot be completely eliminated, indeed. The individual characteristics of people 241

vary, so that the problem of individual differences in reaction is considered one of the 242

most important and complex in psychology. Theoretically speaking, we cannot exclude 243

the influence of some factors, such as being in a university auditorium and receiving a 244

notebook could be associated with taking an exam. 245

However, in our experiment, there was no requirement to complete the task in limited time. According to the experimental setting, the subjects were not required 248

to demonstrate any intellectual or other capabilities to be compared with their peers. The 249

instructions given to students before the experiment explained that the proposed task 250

had no right or wrong answers, the drawing directions are the matter of their personal 251

choice that would neither be evaluated by the experimenter, nor by other subjects, nor by 252

third parties, in any way. The participants were also notified that their work would not be 253

compared with the results of other subjects, and all the information obtained in the study 254

would be used only for the anonymized computer analysis. No time limits were set in the 255

experiment. The experiment began after all subjects confirmed that they understood the 256

instructions and were ready to complete the task. The experiment ended when all subjects 257

completed the task proposed to them. The experiment was carried out in small groups in 258

comfortable, well-lit classrooms. The detailed plan of the study was published and widely 259

announced among the students in the Institute of Psychology well before the enrollment. 260

The study did not involve randomization and group classification of participants in the 261

context of intervention under assessment. Each and every participant freely and deliberately 262

read, agreed, and signed the participation consent form. Therefore, we have no reason to 263

believe that the students participating in the experiment were in conditions of anxiety or 264

depression, as they showed a personal interest in taking part in a psychological experiment, 265

familiarized themselves with the experimental setting, and agreed to participate in the 266

study. 267

In regard to the nature and structure of the experimental data collected, it is worth 268

mentioning that rather than using statistical methods consecrated by tradition, in our work, 269

we implemented machine learning methods (explained in the forthcoming section in detail) 270

to find generalizable predictive patterns and clusters in the data. While statistics draws 271

population inferences from a sample through the fitting of some hypothesized probability 272

distribution to the data, machine learning makes minimal assumptions about the data- 273

generating systems, and its general-purpose learning algorithms retain their high efficiency 274

even when the data are gathered without a carefully controlled experimental design and 275

in the presence of strong nonlinear interactions [54]. The high individual sensation differences and excessive variability in personal sensitivity traits of subjects participated in 277

our experiment made the use of standard statistical methods ineffective. Initial attempts 278

at usual statistical data processing undertaken immediately after the experiments in 2016 were not successful. However, the use of machine learning methods made it possible to 280

classify the collected data.

Gender (sex) differences in behavioral patterns might be explored in the current study

We thank our reviewer for the suggestion! The following paragraph has been added to the results section:

Observed behavior differences are not related to gender. Men and women were equally presented in each group. In Fig. 4(a), we have shown a scat- 391

ter plot representing the male and female participants in the feature space of experimental 392

data collected in the first Experiment by black and white circles, respectively. In Fig. 4(b), 393

we juxtapose two color-coded kernel density plots of the preferred line drawing angles 394

observed in Experiment 1 for the male and female subjects. Two humps clearly visible 395

in Fig. 4(b) for both male and female participants indicate the common, sex irrelevant 396

preference to draw lines in the upper-right and lower-left quadrants.

Figure 4. (a) A scatter plot representing the male and female participants in the feature space of

experimental data collected in Experiment 1. (b) The kernel density plots of the line drawing angles

in Experiment 1, for the male and female subjects. The Man-Whitney U-test of the distributions of

line drawing angles shows that there is no statistically significant sex specific difference in line angel

preferences between men and women.

The p-value calculated from the Man-Whitney U-test used to compare the sex specific 398

samples shown in Fig. 4(b) is p = 0.1474 > 0.05 in favor of that there is no statistically 399

significant difference between preferred angles of lines drawn by men and women.

Minor, the outlet of the Abstract might be structured, for a more understandable effect.

Thank you very much! The Abstract has been correct and expanded in the following way:
“Abstract: A psychology experiment on deciding over a continuum of subjectively equivalent 1

alternatives (directions) revealed that subjects follow a common pattern, giving preferences to just a 2

few directions over all others. In the restricted experimental settings making the common pattern 3

unfeasible, subjects demonstrated no common choice preferences. In the latter case, the observed 4

distribution of choices made by a group of subjects was close to normal. We conclude that abundance 5

of subjectively equivalent alternatives may reduce the individual variability of choices, and vice 6

versa. Choice overload paradoxically results in behavior patterning and eventually facilitates decision 7

predictability while restricting the range of available options fosters individual variability of choice 8

reflected in almost random behavior over the group.

We greatly appreciate time and efforts of our reviewers helping us to improve the quality of our manuscript!

Round 2

Reviewer 2 Report

I think the quality of article has been improved greatly, and it can be accepted for publication. Many thanks for the efforts of authors.

Reviewer 3 Report

No further comments